

# Supplementation of banana peel powder for the development of functional broiler nuggets

Tasleem Akram[1], Sharmeen Mustafa[1], Khola Ilyas[1], Muhammad Rizwan Tariq[2], Shinawar Waseem Ali[2], Sajid Ali[3], Muhammad Shafiq[4], Maryam Rao[2], Waseem Safdar[5], Madiha Iftikhar[6], Amna Hameed[6], Mujahid Manzoor[7], Madiha Akhtar[8], Zujaja Umer[2] and Zunaira Basharat[2]

[1] Faisalabad Medical University, Faisalabad, Punjab, Pakistan
[2] Department of Food Sciences, University of the Punjab, Lahore, Punjab, Pakistan
[3] Department of Agronomy, University of the Punjab, Lahore, Punjab, Pakistan
[4] Department of Horticulture, University of the Punjab, Lahore, Punjab, Pakistan
[5] Department of Biological Sciences, National University of Medical Sciences, Rawalpindi, Punjab, Pakistan
[6] Department of Diet and Nutritional Sciences, Ibadat International University, Islamabad, Pakistan
[7] Department of Entomology, University of the Punjab, Lahore, Punjab, Pakistan
[8] Punjab Food Authority, Rawalpindi, Punjab, Pakistan

Corresponding authors
Muhammad Rizwan Tariq,
rizwan.foodsciences@pu.edu.pk
Maryam Rao,
maryamrao982@gmail.com

## ABSTRACT

Banana peel powder is considered one of the most nutritive and effective waste product to be utilized as a functional additive in the food industry. This study aimed to determine the impact of banana peel powder at concentrations of 2%, 4%, and 6% on the nutritional composition, physicochemical parameters, antioxidant potential, cooking properties, microbial count, and organoleptic properties of functional nuggets during storage at refrigeration temperature for 21 days. Results showed a significant increase in nutritional content including ash and crude fiber ranging from $2.52 \pm 0.017\%$ to $6.45 \pm 0.01\%$ and $0.51 \pm 0.01\%$ to $2.13 \pm 0.01\%$, respectively, whereas a significant decrease was observed in crude protein and crude fat ranging from $13.71 \pm 0.02\%$ to $8.92 \pm 0.02\%$ and $9.25 \pm 0.02\%$ to $4.51 \pm 0.01\%$, respectively. The incorporation of banana peel powder significantly improved the Water Holding Capacity from 5.17% to 8.37%, cooking yield from $83.20 \pm 0.20\%$ to $87.73 \pm 0.16\%$ and cooking loss from $20.19 \pm 0.290\%$ to $13.98 \pm 0.15\%$. Antioxidant potential was significantly improved as TPC of functional nuggets increased ranging from $3.73 \pm 0.02$ mg GAE/g to $8.53 \pm 0.02$ mg GAE/g while a decrease in TBARS ($0.18 \pm 0.02$ mg malonaldehyde/kg to $0.14 \pm 0.02$ mg malonaldehyde/kg) was observed. Furthermore, functional broiler nuggets depicted a significantly reduced total plate count ($3.06$–$4.20 \times 10^5$ CFU/g) than control, which is likely due to high amounts of phenolic compounds in BPP. Broiler nuggets supplemented with 2% BPP (T1) received the greatest sensory scores in terms of flavour, tenderness, and juiciness. Results of current study revealed the potential of BPP to be utilized as an effective natural source of fibre supplementation in food products along with enhanced antioxidant and anti-microbial properties.

# INTRODUCTION

The needs of consumers in the zone of food production have changed significantly during the last few years. Nowadays, foods are intended not only to satisfy hunger and offer essential nutrients for human beings but also to cure nutritional ailments and improve physical and mental health (*Siró et al., 2008*). Functional foods have been categorized as industrially treated or natural foodstuffs; when ingested in a varied diet at adequate levels may provide valuable health benefits apart from essential nutrition (*Alongi & Anese, 2021*). The banana fruit (*Musa balbisiana*) is a tropical and subtropical crop that is the most important fruit crop after rice crop, wheat crop, and maize globally. Over the past 20 years, banana production has steadily increased, going from over 70 million tonnes in 1999 to nearly 117 million in 2019 (*Zaini et al., 2022*). The banana peel accounts for 35–50% of the fruit overall. Enormous amount of banana peels are wasted daily in fruit markets and household garbage, leading to environmental pollution (*Gomes et al., 2020*).

Banana peel is an excellent reservoir of macronutrients, micronutrients, and many bioactive compounds. Banana peel is abundant in total dietary fiber, crude protein, carbohydrates (cellulose, hemicellulose, pectin, and lignin), minerals (calcium, magnesium, phosphorus, and potassium), amino acids (leucine, threonine valine, and phenylalanine), and polyunsaturated fatty acids, particularly linoleic acid (*Azarudeen & Nithya, 2021*). Banana peel is one of the most important waste products that might be employed in the food sector as a functional additive (*Zaini et al., 2020a*). Bioactive substances found in banana peels include phlobatannins, tannins, alkaloids, flavonoids, glycosides, anthocyanins, and terpenoids, all of which have biological and medicinal properties such as antibacterial, anti-inflammatory, antidiabetic, and anti-hypertensive properties (*Pereira & Maraschin, 2015*). The bioactive chemicals present in banana peel provide useful medicinal properties, including immunostimulant effects, to the fruit peel. A sufficient amount of antioxidants such as polyphenols, carotenoids, catecholamines, and prodelphinidins are present in the banana peel making it a functional additive in food products associated with anti-oxidative and anti-microbial properties (*Vu, Scarlett & Vuong, 2018*).

Dietary fiber is a non-starch polysaccharide having resistance to absorption and digestion by the enzymes present in the human gastrointestinal tract. Since the human body cannot digest and absorb dietary fiber, moisture absorption in the digestive system is affected (*Yang et al., 2017*). Fiber should not only afford health benefits but also provide enhanced technological properties to encourage continued high-fiber product intake. Fiber is ideal for meat product development due to its water retention property, neutral flavor, and reduces cooking loss (*Zaini, Sintang & Pindi, 2020b*).

Nutritionally, meat is a good source of high-value protein and essential fatty acids, vitamins, and minerals (*Ahmad, Imran & Hussain, 2018*). Among all meat types, broiler meat consumption has expanded dramatically over the previous several decades and seems to expand in the future (*Petracci et al., 2013*). Broiler meat is a low-calorie food with a high

nutrient density that is a significant source of necessary polyunsaturated fatty acids (PUFAs), notably omega-3 fatty acids, protein, vitamins, and minerals. Furthermore, poultry meat has a neutral flavor, nice stable texture, and light color, making it more suitable for processing than other meat types (*Barbut, 2012*). Despite its nutritional value, chicken has several disadvantages, such as a lack of dietary fiber and a higher risk of lipid oxidation attributed to a high concentration of polyunsaturated fatty acid that causes changes in meat colour, flavours, texture, and nutritional value (*Das et al., 2020*). Though meat products are a primary source of several nutrients, they have also been associated with adverse health effects like diabetes, high cholesterol, obesity, cardiovascular, and other diseases. More efforts are being made to produce "healthier meat products" by reducing unhealthy components like saturated fats, salt, and nitrates and stabilize it by increasing its antioxidant potential without disrupting the nutritional profile (*Mora-Gallego et al., 2016*). The present study was designed to assess the impact of supplementation of banana peel powder for the development of functional broiler nuggets with the objectives to prepare low-fat nuggets by addition of banana peel powder, determining the nutritional and chemical characteristics of nuggets and analyzing storage stability and sensory attributes of functional nuggets.

## MATERIALS AND METHODS

### Procurement of raw materials

The bananas were procured from the local market of Lahore, Pakistan. Boneless fresh broilers meat was purchased from the local groceries store in Lahore, Pakistan. The meat was packed in small bags of LDPE and held in a refrigerator at (4 ± 1 °C), which later was used for functional nuggets formulation. The other ingredients used in nuggets preparation, *i.e.*, bread, garlic powder, ginger powder, white pepper, soya sauce, iodized salt, and all chemicals and solvents used in this experiment were of analytical grade purchased from Descon OxyChem Ltd. (Lahore, Pakistan) and ICI Pakistan Ltd. (Karachi, Pakistan).

### Preparation of banana peels powder (BPP)

The purchased bananas were washed with water to eliminate dust and debris. The pulp and peel of the banana fruit were separated. Banana peels were cut into small pieces and treated with a 0.5% (w/v) solution of citric acid for 10 min to prevent enzymatic browning. The solution of citric acid was drained and cut pieces of peels of bananas were dried using a hot air oven (DHG-9053A, Thermomatic, India) at a temperature of 40 °C for 48 h. After that, the banana peels were dried, ground, and then screened using 60 mesh (250 μm) screens, resulting in banana peel powder (*Agama-Acevedo et al., 2009*). The peel powder was then stored in airtight plastic packs and was kept in a cold place for compositional analysis, determination of anti-oxidant potential, and product development.

### Compositional study of banana peel powder

Proximate parameters of banana peel powder, such as moisture, crude protein, crude fat, total ash, and crude fiber were determined as per standard protocols of *AOAC (2005)*.

**Table 1 Formulation of broiler meat nuggets with banana peel powder.**

| Ingredients (g) | Control (T0) | Treatment I (T1) | Treatment II (T2) | Treatment III (T3) |
|---|---|---|---|---|
| Broiler meat | 70 | 68 | 66 | 64 |
| Banana peel powder | 0 | 2 | 4 | 6 |
| Bread | 22 | 22 | 22 | 22 |
| Garlic powder | 1 | 1 | 1 | 1 |
| Ginger powder | 1 | 1 | 1 | 1 |
| White pepper | 2 | 2 | 2 | 2 |
| Soya sauce | 2 | 2 | 2 | 2 |
| Salt | 1 | 1 | 1 | 1 |

**Note:**
Control, nuggets without BPP; T1, nuggets with 2% BPP; T2, nuggets with 4% BPP; T3, nuggets with 6% BPP; bread, garlic powder, ginger powder, white pepper, soya sauce, and salt were kept constant in all preparation.

The sample was oven-dried (65 °C for 120 min), and moisture content was measured as weight difference. The Kjeldahl method was used for the estimation of crude protein. The crude fat percentage was obtained using the solvent extraction method using methanol as solvent. The total ash content of banana peel powder was determined using a muffle furnace at 550 °C. The crude fiber was determined by the digestion of banana peel powder with acid and alkali. Analyses of all the samples were performed in triplicates.

The carbohydrate (%) of banana peel powder was estimated as described by *Romelle, Rani & Manohar (2016)* by using the formulae:

$$\text{Carbohydrate }(\%) = 100 - (\text{moisture \%} + \text{protein \%} + \text{crude fat \%} + \text{ash \%} + \text{crude fiber \%})$$

The antioxidant potential of banana peel powder was measured by DPPH (2,2-diphenyl-1-picrylhydrazyl) assay following the method of (*Durgadevi, Saravanan & Uma, 2019*), using methanol extracts. Absorbance was measured by spectrophotometer (NIR spectrophotometer BK-S430, Japan) at 515 nm at the end of the experiment against a control, both the control and sample values were recorded. The total phenolic content of banana peel powder was estimated using the Folin-Ciocalteu method (*Özünlü, Ergezer & Gökçe, 2018*), and the results were measured in mg GAE per 100 g of banana peel powder.

## Development of functional broiler nuggets

Minced broiler meat and bread were mixed along with spices garlic powder, ginger powder, salt, white pepper, and soya sauce according to the formulation mentioned in Table 1. After mixing, banana peel powder was added in a concentration of 0%, 2%, 4%, and 6%, and treatments were named T0, T1, T2, and T3, respectively. The mixture was thoroughly stirred until it reached the appropriate texture. A homogeneous size of 3 cm × 2 cm × 1 cm was achieved by shaping the mixture of nuggets. After the nuggets had been shaped to the desired size, they were coated with egg and breadcrumbs. The functional nuggets were packaged aerobically in low-density polyethylene (LDPE) boxes and stored in the refrigerator to determine physicochemical attributes, and storage stability (0, 7, 14, and

21 days). The product was air fried at 200 °C for 10 min to evaluate cooking and sensory attributes.

## Physicochemical analysis of functional nuggets

The proximate composition (moisture, crude fat, crude protein, crude fiber, and total ash) was analyzed using standard protocols (*AOAC, 2005*). The carbohydrate (%) of broiler nugget was estimated by taking the difference of all proximate parameters (*Romelle, Rani & Manohar, 2016*). Water holding capacity (WHC) was determined following the method stated by *Jin et al. (2007)*. The sample (5 g) was centrifuged at 5 °C for 15 min. The pH of the samples was determined using a pH meter (pH meter S400 basic, Mettler-Toledo, Columbus, OH, USA) according to *Manigiri et al. (2019)*. Analyses of all the samples were performed in triplicates on days 0, 7, 14, and 21. Total phenolic content was estimated using the Folin-Ciocalteu method as described by *Özünlü, Ergezer & Gökçe (2018)* with few modifications. Broiler nuggets (1 g) were homogenized in 10 ml methanol and kept overnight at 4 °C for extraction. 2.5 ml of FC reagent and 7.5% sodium carbonate (2 mL) were added in aliquots of 0.5 ml broiler nugget extracts and rested at room temperature for 30 min. Absorbance was measured at 760 nm as milligrams of gallic acid equivalents (GAE) per 100 gram of nugget. Sample (5 g) was mixed with trichloroacetic acid (20 ml, 0.5 M) and homogenized for 2 min, followed by filtration. Thiobarbituric acid (TBA) solution (2 mL) was added to the filtrate (2 mL) and incubated at 95 °C for 35 min. The absorbance of the sample was detected at 532 nm using a spectrophotometer as mg of malonaldehyde per kg (*Zaini, Sintang & Pindi, 2020b*).

## Determination of cooking characteristics of functional nuggets

The cooking yield was calculated by subtracting the weight of cooked nuggets from the weight of raw nuggets according to the method provided by *Pathera et al. (2017)*. Following the method developed by *Haque et al. (2020)*, cooking loss was estimated by dividing the weight difference between cooked and uncooked nuggets by the weight of the food before cooking.

## Microbiological analysis of functional nuggets

Microbial load (total plate count) of all treatments were evaluated by using method of *Salazar et al. (2021)* with few modifications. Media was prepared and autoclaved at 15 psi pressure at 121 °C for 15 min. Dilutions were made by adding 1 g of nugget sample to 9 ml peptone water, and up to five dilutions had made. Inoculation was done by streak plate method and plates were incubated for two days at 37 °C. The results were expressed as colony-forming units per g (CFU $\times 10^5$/g) of the sample.

## Sensory evaluation

After air frying, five trained panellists assessed each treatment of broiler nuggets enriched with banana peel powder for organoleptic qualities. The test was conducted using a 9-point hedonic scale, where nine stood for strongly like and one for extremely dislike (*De-Carvalho, da Silva & Giada, 2018*). Sensory attributes (colour, flavour, tenderness,

**Table 2 Nutritional composition of banana peel powder.** Proximate composition and antioxidant potential.

| Parameter | Analysis | Results |
|---|---|---|
| Proximate composition | Moisture (%) | 6.80 ± 1.92 |
| | Crude protein (%) | 4.01 ± 0.96 |
| | Crude fat (%) | 3.40 ± 0.40 |
| | Ash (%) | 11.04 ± 1.39 |
| | Crude fiber (%) | 16.66 ± 0.57 |
| | Carbohydrates (%) | 61.45 ± 1.82 |
| Antioxidant potential | TPC (mg GAE/g) | 55.53 ± 0.49 |
| | DPPH (%) | 76.03 ± 0.10 |

juiciness, and overall acceptability) were evaluated after every 7 days till 21 days storage period.

## Statistical analysis

Physicochemical and sensory properties of broiler nuggets were examined in relation to the impacts of adding banana peel powder using descriptive statistics (means, standard deviation (SD)), t-test, Analysis of Variance (ANOVA). Version 25.0 of Statistical Package for the Social Sciences (SPSS) software was used to analyse the data. For multiple comparisons, two-way ANOVA and least significant difference (LSDs) *post hoc* analysis were performed. *p*-values of ($p < 0.05$) were regarded as statistically significant for all tests.

## RESULTS

### Compositional study of banana peel powder

Nutritional composition of hot air oven-dried banana peel powder is presented in Table 2. Moisture content in banana peel powder was 6.80% ± 1.92. Significant crude protein (4.01% ± 0.96) content was also recorded. Considerably low (3.40 ± 0.40%) fat content was observed in banana peel powder. The ash percentage found in banana peel powder was 11.04% ± 1.39. Crude fiber and carbohydrates was 16.66% ± 0.57 and 61.45% ± 1.82 respectively. Total phenolic content observed in banana peel powder was 55.53 ± 0.49 mg GAE/g. High amount of total phenolic content leads to improved radical scavenging ability (76.03% ± 0.10) determined by DPPH (2,2-diphenyl-1-picrylhydrazyl).

### Proximate composition of functional nuggets

The results of the physicochemical analysis of all treatments are depicted in Table 3. The incorporation of banana peel powder significantly reduced the moisture content ranging from (70.51 ± 0.49% to 67.60 ± 0.64%) of nuggets during refrigerated storage. The decreased protein content of nuggets ranged from 13.71 ± 0.01% to 8.92 ± 0.01%. Fat percentage was observed significantly lower for broiler nuggets supplemented with the banana peel powder as compared to the control sample and reduced to 3.40%. The inclusion of BPP in nuggets significantly increased the ash content from 2.51 ± 0.03% to 6.44 ± 0.01% with the addition of banana peel powder. Furthermore, the addition of BPP

Table 3 Proximate composition (g/100 g) of functional broiler nugget samples supplemented with banana peel powder.

| Proximate composition | Nugget samples | | | |
|---|---|---|---|---|
| | Control | T1 | T2 | T3 |
| **Moisture** | | | | |
| 0 day | $70.51^a \pm 0.09$ | $69.70^{bc} \pm 0.07$ | $68.69^{cd} \pm 0.07$ | $67.60^{de} \pm 0.06$ |
| 7 day | $68.70^{ab} \pm 0.06$ | $67.84^{cd} \pm 0.01$ | $66.02^{def} \pm 0.03$ | $65.24^{fg} \pm 0.03$ |
| 14 day | $67.87^{bc} \pm 0.06$ | $66.77^{efg} \pm 0.04$ | $65.69^{efg} \pm 0.03$ | $64.99^{gh} \pm 0.08$ |
| 21 day | $66.88^{fg} \pm 0.07$ | $65.65^g \pm 0.04$ | $64.87^{gh} \pm 0.04$ | $63.96^h \pm 0.06$ |
| **Crude protein** | | | | |
| 0 day | $13.71^a \pm 0.07$ | $12.81^b \pm 0.05$ | $10.12^f \pm 0.05$ | $8.92^i \pm 0.01$ |
| 7 day | $12.62^b \pm 0.05$ | $11.77^d \pm 0.01$ | $9.21^h \pm 0.01$ | $7.83^k \pm 0.01$ |
| 14 day | $11.71^d \pm 0.05$ | $10.72^e \pm 0.01$ | $8.11^j \pm 0.01$ | $6.83^m \pm 0.05$ |
| 21 day | $10.64^e \pm 0.05$ | $9.67^g \pm 0.05$ | $7.18^l \pm 0.02$ | $5.91^n \pm 0.01$ |
| **Crude fat** | | | | |
| 0 day | $9.25^a \pm 0.01$ | $7.35^b \pm 0.01$ | $5.60^c \pm 0.02$ | $4.51^d \pm 0.01$ |
| 7 day | $9.25^a \pm 0.01$ | $7.33^b \pm 0.01$ | $5.58^c \pm 0.01$ | $4.50^d \pm 0.01$ |
| 14 day | $9.24^a \pm 0.01$ | $7.34^b \pm 0.04$ | $5.59^c \pm 0.01$ | $4.50^d \pm 0.01$ |
| 21 day | $9.22^a \pm 0.01$ | $7.33^b \pm 0.01$ | $5.57^c \pm 0.01$ | $4.49^d \pm 0.01$ |
| **Ash** | | | | |
| 0 day | $2.51^d \pm 0.02$ | $4.34^c \pm 0.06$ | $5.54^b \pm 0.05$ | $6.44^a \pm 0.01$ |
| 7 day | $2.52^d \pm 0.02$ | $4.36^c \pm 0.04$ | $5.56^b \pm 0.03$ | $6.45^a \pm 0.01$ |
| 14 day | $2.51^d \pm 0.05$ | $4.36^c \pm 0.05$ | $5.57^b \pm 0.01$ | $6.44^a \pm 0.05$ |
| 21 day | $2.52^d \pm 0.05$ | $4.36^c \pm 0.05$ | $5.57^b \pm 0.05$ | $6.45^a \pm 0.05$ |
| **Crude fibre** | | | | |
| 0 day | $0.51^d \pm 0.05$ | $0.96^c \pm 0.05$ | $1.62^b \pm 0.02$ | $2.12^a \pm 0.05$ |
| 7 day | $0.51^d \pm 0.01$ | $0.96^c \pm 0.02$ | $1.61^b \pm 0.01$ | $2.11^a \pm 0.05$ |
| 14 day | $0.50^d \pm 0.05$ | $0.95^c \pm 0.01$ | $1.60^b \pm 0.01$ | $2.10^a \pm 0.05$ |
| 21 day | $0.50^d \pm 0.05$ | $0.95^c \pm 0.05$ | $1.60^b \pm 0.05$ | $2.10^a \pm 0.10$ |
| **Carbohydrates** | | | | |
| 0 day | $3.50^i \pm 0.05$ | $4.81^h \pm 0.09$ | $8.41^f \pm 0.07$ | $10.39^e \pm 0.06$ |
| 7 day | $6.39^g \pm 0.06$ | $7.71^f \pm 0.06$ | $11.99^d \pm 0.03$ | $13.84^c \pm 0.03$ |
| 14 day | $8.15^f \pm 0.07$ | $9.84^e \pm 0.04$ | $13.42^c \pm 0.03$ | $15.11^b \pm 0.08$ |
| 21 day | $10.21^e \pm 0.06$ | $12.02^d \pm 0.04$ | $15.18^b \pm 0.03$ | $17.07^a \pm 0.07$ |

Note:
Mean values ± SD with different letters (a–n) in individual parameters (4 rows each) are significantly different ($p < 0.05$). Control, without BPP; T1, nuggets with 2% BPP; T2, nuggets with 4% BPP; T3, nuggets with 6% BPP.

considerably enhanced the crude fiber content which ranged from 0.51 ± 0.01% to 2.13 ± 0.01 in treated samples as compared to the control. Carbohydrates of treated samples were improved significantly and were found significantly higher in samples treated with banana peel powder than in the control sample.

## Physiochemical parameters of functional nuggets

Physicochemical parameters of functional nuggets containing different concentrations of banana peel powder given as 2%, 4%, and 6% are depicted in Table 4. The antioxidant potential of the banana peel powder-supplemented functional nuggets is provided in Fig. 1. The TBARS values observed for functional nuggets supplemented with banana peel powder are provided in Fig. 2. The water-holding capacity of nuggets was improved significantly ($p < 0.05$) and was observed to be in the range of 6.71 ± 0.017% to 9.92 ± 0.017%. Analysis of pH indicated that pH values of nuggets supplemented with banana

**Table 4 Physicochemical parameters of functional broiler nugget samples supplemented with banana peel powder.**

| Physicochemical parameters | Nugget samples | | | |
|---|---|---|---|---|
| | T0 | T1 | T2 | T3 |
| Water holding capacity (%) | | | | |
| 0 day | 6.71$^g$ ± 0.01 | 7.81$^d$ ± 0.01 | 8.12$^c$ ± 0.02 | 9.92$^a$ ± 0.01 |
| 7 day | 5.62$^i$ ± 0.01 | 6.77$^g$ ± 0.01 | 7.21$^e$ ± 0.01 | 8.83$^b$ ± 0.01 |
| 14 day | 4.71$^k$ ± 0.01 | 5.72$^i$ ± 0.04 | 6.11$^h$ ± 0.01 | 7.83$^d$ ± 0.05 |
| 21 day | 3.64$^l$ ± 0.01 | 4.67$^k$ ± 0.01 | 5.18$^j$ ± 0.02 | 6.91$^f$ ± 0.01 |
| pH | | | | |
| 0 day | 5.95$^a$ ± 0.05 | 5.96$^a$ ± 0.01 | 5.94$^a$ ± 0.05 | 5.97$^a$ ± 0.05 |
| 7 day | 5.78$^b$ ± 0.01 | 5.79$^b$ ± 0.02 | 5.67$^c$ ± 0.02 | 5.58$^d$ ± 0.05 |
| 14 day | 5.46$^e$ ± 0.05 | 5.70$^c$ ± 0.05 | 5.36$^f$ ± 0.01 | 5.45$^e$ ± 0.05 |
| 21 day | 5.40$^f$ ± 0.05 | 5.60$^d$ ± 0.05 | 5.20$^h$ ± 0.05 | 5.31$^g$ ± 0.01 |

Note:
Mean values ± SD with different letters (a–l) in individual parameters (4 rows each) are significantly different ($p < 0.05$). Control, nuggets without BPP; T1, nuggets with 2% BPP; T2, nuggets with 4% BPP; T3, nuggets with 6% BPP.

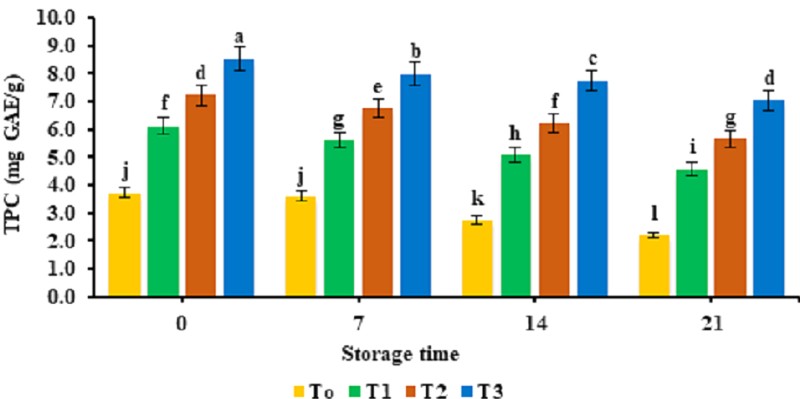

**Figure 1  Total phenolic content.** Total phenolic content TPC (mg GAE/g) of functional broiler nugget samples supplemented with banana peel powder.      

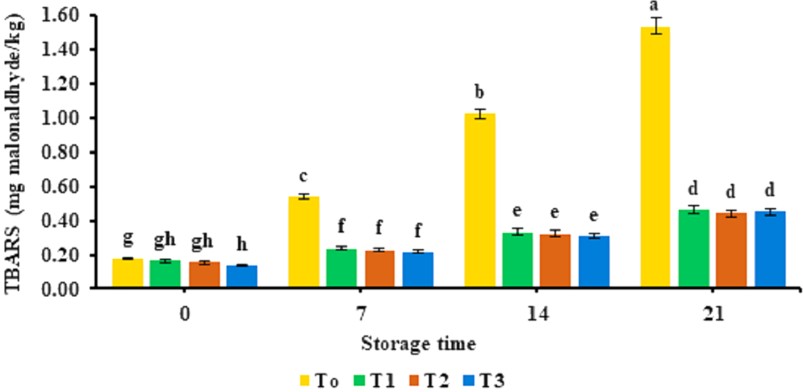

**Figure 2  Thiobarbituric acid reactive substances.** Thiobarbituric acid reactive substances values TBARS (mg malonaldhyde/kg) in functional broiler nugget samples supplemented with banana peel powder.      

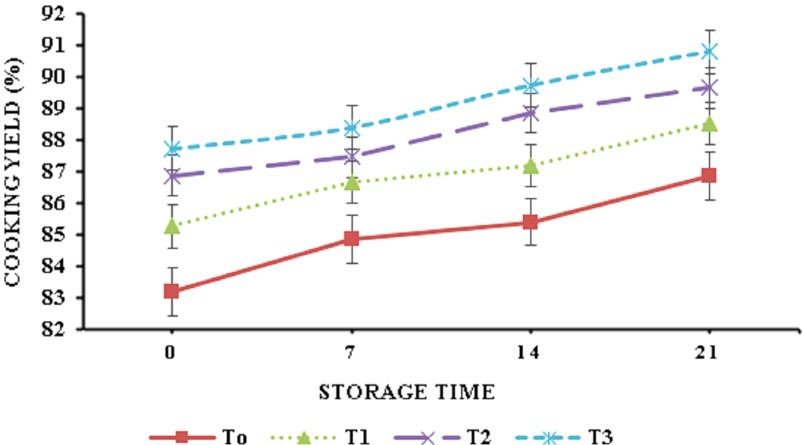

**Figure 3 Cooking yield.** Cooking yield (%) of functional broiler nugget samples supplemented with banana peel powder.  

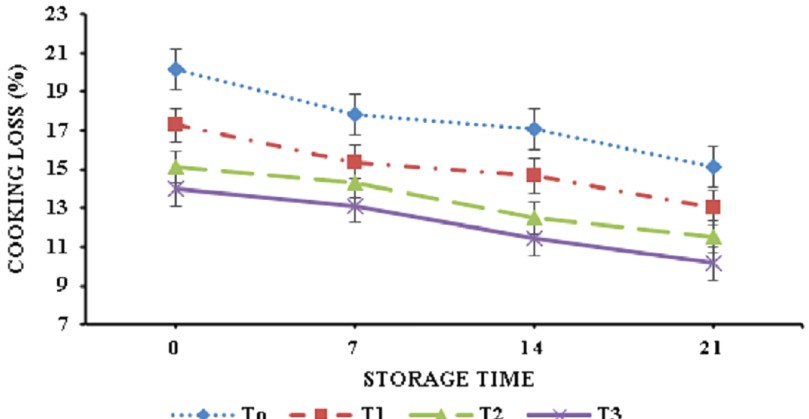

**Figure 4 Cooking loss.** Cooking loss (%) of functional broiler nugget samples supplemented with banana peel powder.  

peel powder were decreased to a statistically significant level ($p < 0.05$). An increase in total phenolic content was observed in functional nuggets (3.73 ± 0.017 mg GAE/g to 8.53 ± 0.017 mg GAE/g) as compared to banana peel powder. The highest antioxidant activity was observed in the sample with 6% banana peel powder. However, a significant decrease in the total phenolic content was observed with the storage time of 21 days and the least TPC was observed at day 21. Analysis for TBARS values indicated that it decreased significantly ($p < 0.05$) with the addition of banana peel powder. The least values of TBARS were observed for the T3 sample supplemented with 6% banana peel powder.

## Cooking characteristics of functional nuggets

The data regarding cooking yield and cooking loss is depicted in Figs. 3 and 4. Banana peel powder had a highly significant effect ($p < 0.05$) on the cooking attributes of broiler meat nuggets. Results showed that cooking yield of nuggets varied from 85.08–89.16% during storage. T3 (6% BPP) showed highest cooking yield (90.80%) on day 21 and the control
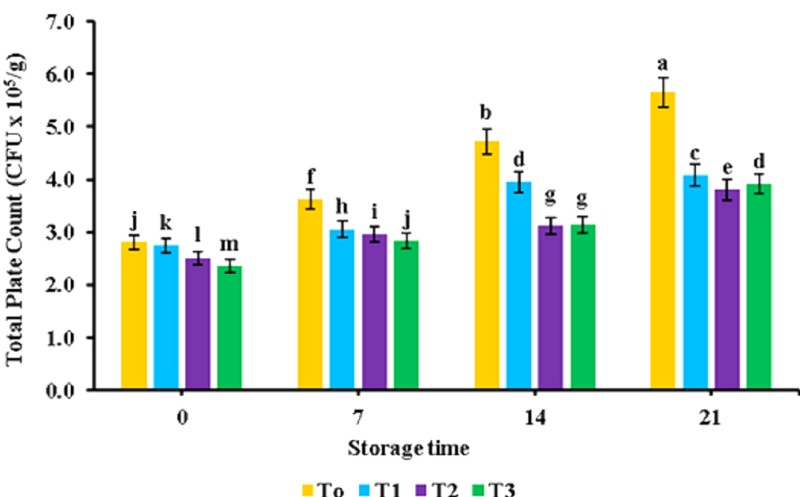

**Figure 5 Total plate count.** Total plate count (cfu $\times 10^5$/g) of functional broiler nugget samples supplemented with banana peel powder.

sample (T0) 0% BPP presented the lowest cooking yield (83.20%) at 0 day. A significant decrease ($p < 0.05$) was observed in cooking loss and it ranged from $20.19 \pm 0.29$% to $13.98 \pm 0.15$% for nuggets supplemented with BPP in comparison to the control. Physiochemical analysis revealed that treatment containing 6% BPP (T3) showed best results at day 21 as compared to control and all other treatments.

## Microbiological analysis

The bacterial load of broiler nugget samples supplemented with banana peel powder is displayed in Fig. 5. The lower microbial count ($2.36 \pm$ CFU $\times 10^5$/g) was detected in nuggets having banana peel powder as compared to the control ($2.81 \pm$ CFU $\times 10^5$/g). The maximum microbial load (4.20 CFU $\times 10^5$/g) was observed in the control treatment without banana peel powder while the lowest total plate count (3.06 CFU $\times 10^5$/g) was observed in T3. During the storage time of 21 days, the bacterial load of nugget samples depicted a significant increase ($p < 0.05$).

## Sensory evaluation

The sensory scoring for different organoleptic attributes of control and treated samples is illustrated in Fig. 6. The addition of banana peel powder significantly affected the sensory parameters of treated samples. The colour score slightly decreased in broiler nuggets after increasing the concentration of banana peel powder. An elevated concentration of banana peel powder decreased flavor of broiler nuggets. A concentration of banana peel powder above 2% significantly affected tenderness and contributed to the hardness of functional nuggets. Broiler nuggets containing 2% banana peel powder (T1) had a superior sensory acceptance concerning the juiciness. Functional broiler nuggets supplemented with 2% banana peel powder (T1) were regarded as the best treatment based on the highest overall acceptability scores during the whole storage period.

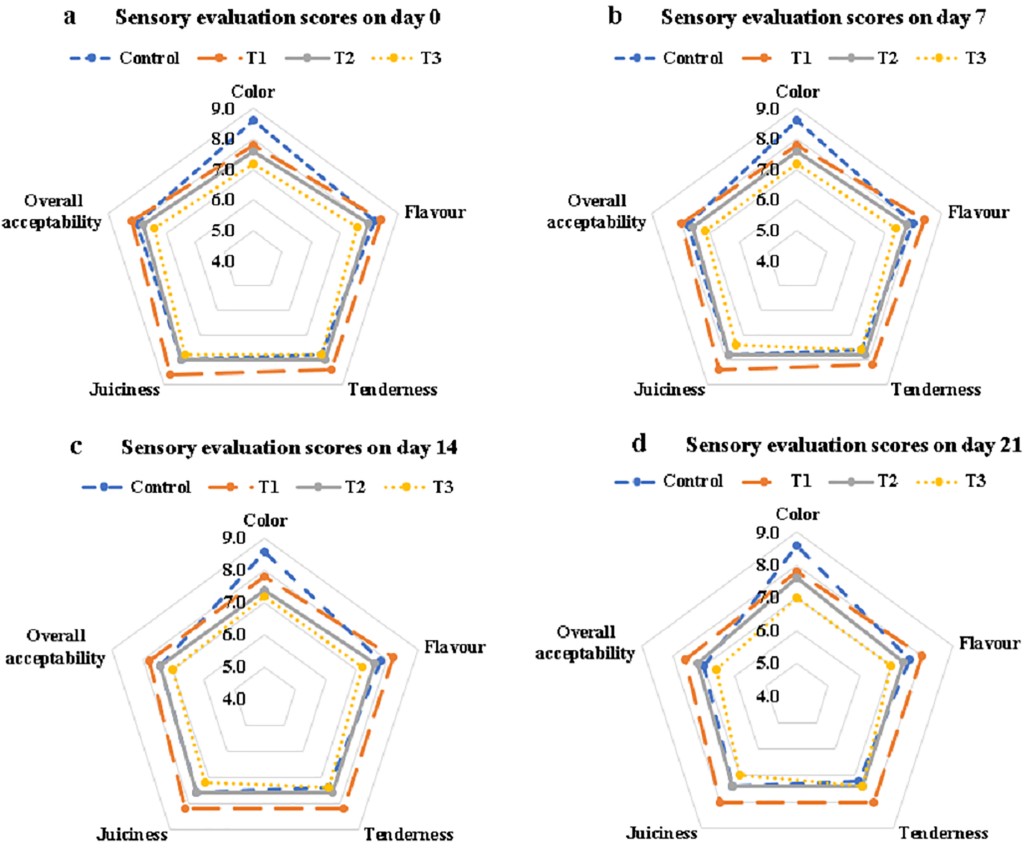

**Figure 6 Sensory evaluation score of functional nuggets.** Sensory evaluation score on day 0 (A); sensory evaluation score on day 7 (B); sensory evaluation score on day 14 (C); sensory evaluation score on day 21 (D).    

## DISCUSSION

The moisture content observed in banana peel powder was in line with *Eshak (2016)* who reported banana peel powder was used as a functional ingredient to improve the nutritional profile of flatbread. Crude protein content revealed on analysis of banana peel powder was comparable to *Lee et al. (2010)* who stated that banana peel powder is a good source of amino acids. Fat content was found to be lower than (13.1% ± 0.2) the results by results observed by *Wachirasiri, Julakarangka & Wanlapa (2009)*. This might be due to the differences in geographical regions or varieties of banana. Similar results were obtained by *Emaga et al. (2007)* for the ash content of banana peel powder who indicated that the content of ash ranged from 6.4 to 12.8% in banana peels. Being high in mineral contents banana peel powder can be utilized as a novel source for enhancing the nutritional profiles of different food products. Crude fiber and carbohydrate contents in banana peel powder were found significant and the results are in accordance with *Zaini et al. (2020a)* who analyzed physicochemical characteristics of banana peel powder and its effects on fish patties. Total phenolic contents were found significantly higher than the stated results of *Rebello et al. (2014)* who estimated 29 mg GAE/g total phenolic content in the banana peel. The variation of TPC in banana peels might be influenced by variety, growth environment,

harvesting time, sample preparation, and methods of determination (*Islam et al., 2020*). The DPPH radical scavenging activity depicted results that agree with *Rebello et al. (2014)* and *González-Montelongo, Lobo & González (2010)*, who proposed that several phenolic acids, including hydroxycinnamic acids, flavonoids like dopamine, L-dopa, quercetin, catecholamines, as well as anthocyanins are the main polyphenols in banana peels that are responsible for its high antioxidant potential.

Similar findings were made for moisture analysis by *Yadav et al. (2018)* and *Kaur, Kumar & Bhat (2015)* who discovered that adding pomegranate seed powder and tomato powder to fiber-enriched chicken nuggets reduced their moisture content. Reduced protein contents were attributed to low levels of crude protein in banana peel powder (4.01%). A similar decreasing trend of protein content was observed by *Mahmoud, Abou-Arab & Abu-Salem (2017)* who used orange peel powder in the preparation of beef burgers and reported that protein content of beef patties was reduced by the supplementation of orange peel powder. Fat content decreased because of the low-fat content in banana peel powder. Similar findings were reported by *Zaini et al. (2020a)* who used banana peel powder in fish patties and reported that fat content of patties was reduced and improved the shelf life stability of the fish patties. The ash content improved significantly for nuggets as banana peel powder is a rich source of minerals. *Chappalwar et al. (2021)* prepared chicken patties by the incorporation of banana peel flour and observed a similar effect on the ash content of the product. In another study it was reported that banana peel powder was used to improve the fibre content of waffle cones and results depicted a significant increase in the fibre content of prepared sample (*Zanariah, Zaleqha & Lisnurjannah, 2019*). High crude fiber content can be attributed to high crude fiber in banana peel powder (16.66%). These findings are in line with *Manigiri et al. (2019)* who noted a significant increase in chicken nuggets incorporated with gooseberry seed coat powder. *El-Nashi et al. (2015)* observed an increase in carbohydrates in beef sausages prepared with pomegranate peel powder. An increase in water holding capacity might be attributed to the presence of water molecules residing in empty spaces of fiber in banana peel powder (*Ali, El-Anany & Gaafar, 2011*). These results are in accordance with *Adibelli & Serdaroğlu (2017)* who characterized an increase in the water holding capacity of frankfurters incorporated with 15% apricot pomace. However, a decrease in WHC capacity was observed with the progression of storage for all treatments. The results of pH are in agreement with the findings of *Abidin (2021)* who supplemented fiber in chicken nuggets that lowered pH value and assisted to limit bacterial proliferation by creating an unfavorable environment for microbial growth. Improvement of total phenolic content in nuggets with BPP supplementation associated with the presence of high phenolic content (55.53 mg GAE/g) in banana peel powder. The results of study are consistent with *Sharma & Yadav (2020)* who observed a rise in the total phenolic content of chicken meat patties prepared by the incorporation of pomegranate peel powder. Another study performed by *Arshad et al. (2022)* utilized milk derived bioactive peptides to improve the functional properties of beef nuggets and reported that total phenolic content of nuggets was significantly improved by the supplementation of milk derivatives. However, a decline in TPC of nuggets during storage was observed by *Kabir et al. (2021)* in yogurt who observed a breakdown of the

phenolic compounds due to the presence of lactic acid bacteria during storage time of 28 days at 4 °C. The findings for TBARS values are inconsistent with the outcomes of *Ibrahim & Salem (2013)* who found decreasing trend of TBARS values for the chicken patties formulated with lime peel extract. The banana peels have four times the amount of phenolic content as compared to the pulp which might be responsible for the delayed oxidation of functional broiler nuggets (*Youryon & Supapvanich, 2017*). Furthermore, banana peel comprises high levels of gallocatechin that retain antioxidant and anti-carcinogen potential (*Machado et al., 2017*). Another study performed on the incorporation of sesame seeds in chicken nuggets indicated that the TBARS value of nugget samples reduced significantly by the addition of sesame seeds (*Nawarathne et al., 2021*).

*Zaini et al. (2022)* suggested that the incorporation of banana peel powder in chicken sausages resulted in improved cooking yield which is associated with the water-binding properties of banana peel powder that plays a crucial role in the enhancement of the meat and its product cooking yield (*Agama-Acevedo et al., 2016*). Cooking loss reduction can be attributed to the presence of starch and pectin which interact with proteins to minimize moisture migration during cooking and hence reduced cooking losses (*Zaini et al., 2022*). Similar findings were reported by *Mahmoud, Abou-Arab & Abu-Salem (2017)* where the addition of orange peel powder at a concentration of 2.5%, 5%, 7.5%, and 10% in beef burgers resulted in a decrease in cooking loss of functional burgers. The decrease in total plate count values of nuggets with the supplementation of banana peel powder suggested its antimicrobial potential (*Vu, Scarlett & Vuong, 2018*). *Abidin (2021)* also observed an analogous trend for a microbial load of chicken nuggets formulated with high fiber. *Devendra (2011)* also showed an identical decreasing trend for a microbial load of chicken nuggets enriched with clove powder at a concentration of 0.1%.

Color scores of nuggets supplemented with banana peel powder were reduced because of the dark brown colour of banana peel powder. The taste was perceived as less in the control sample due to the presence of starch which provides a blunt flavor and hard texture. The scoring of sensory evaluation agrees with the findings of *Chappalwar et al. (2021)* and *Zaini, Sintang & Pindi (2020b)* who reported similar effects of banana peel powder and flour on organoleptic characteristics of chicken patties. Contradictory results were reported by another research performed by *Zaini, Sintang & Pindi (2020b)* in which banana peel powder was used to improve the sensorial characteristics of chicken sausages but results depicted that addition of banana peel powder in concentrations of >2% decreased the organoleptic perception of sausage samples. *Rani et al. (2021)* performed a study on chicken meat nugget samples supplemented with pumpkin seed powder and chia seed powder and reported that the sensorial scores of nugget samples were improved by the supplementation of both functional ingredients.

## CONCLUSIONS

Banana peel powder is a good source of fiber and bioactive compounds having possible health-stimulating benefits and enhancing the quality of broiler nuggets. Its supplementation at concentrations of 2%, 4%, and 6% enhances the shelf life of broiler nuggets because of its antimicrobial and antioxidant potential. Proximate parameters of

broiler nuggets such as moisture, crude protein, crude fat, and carbohydrates were significantly affected by the addition of banana peel powder, but ash and crude fiber showed non-significant effects during storage time of 21 days at 4 °C. Furthermore, the incorporation of banana peel powder significantly improved the quality parameters such as WHC, TPC, and cooking characteristics of the product along with considerably reduced lipid oxidation and microbial proliferation during the whole storage time. Sensory evaluation indicated that broiler nuggets made with 2% banana peel powder were most liked by the sensory panel whereas broiler nuggets made with 6% banana peel powder were liked least but in the acceptable score.

### Funding
The authors received no funding for this work.

### Competing Interests
The authors declare that they have no competing interests.

### Author Contributions
- Tasleem Akram conceived and designed the experiments, performed the experiments, prepared figures and/or tables, and approved the final draft.
- Sharmeen Mustafa conceived and designed the experiments, prepared figures and/or tables, and approved the final draft.
- Khola Ilyas conceived and designed the experiments, analyzed the data, authored or reviewed drafts of the article, and approved the final draft.
- Muhammad Rizwan Tariq performed the experiments, analyzed the data, prepared figures and/or tables, and approved the final draft.
- Shinawar Waseem Ali analyzed the data, authored or reviewed drafts of the article, and approved the final draft.
- Sajid Ali analyzed the data, prepared figures and/or tables, and approved the final draft.
- Muhammad Shafiq conceived and designed the experiments, authored or reviewed drafts of the article, and approved the final draft.
- Maryam Rao conceived and designed the experiments, analyzed the data, authored or reviewed drafts of the article, and approved the final draft.
- Waseem Safdar analyzed the data, prepared figures and/or tables, and approved the final draft.
- Madiha Iftikhar performed the experiments, analyzed the data, authored or reviewed drafts of the article, and approved the final draft.
- Amna Hameed performed the experiments, analyzed the data, authored or reviewed drafts of the article, and approved the final draft.
- Mujahid Manzoor conceived and designed the experiments, authored or reviewed drafts of the article, and approved the final draft.

# PeerJ

- Madiha Akhtar performed the experiments, authored or reviewed drafts of the article, and approved the final draft.
- Zujaja Umer conceived and designed the experiments, performed the experiments, prepared figures and/or tables, and approved the final draft.
- Zunaira Basharat conceived and designed the experiments, performed the experiments, prepared figures and/or tables, and approved the final draft.

## Data Availability

The raw data, data of all the parameters in triplicate, and coding for further statistical analysis are available in the Supplemental Files.

## Supplemental Information

Supplemental information for this article can be found online at http://dx.doi.org/10.7717/peerj.14364#supplemental-information.

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
