# Peer review of "Supplementation of banana peel powder for the development of functional broiler nuggets"

_PeerJ, doi:10.7717/peerj.14364_

## Round 0.1 · original submission · Major Revisions

Dear authors, the manuscript submitted for publication consideration has been reviewed by experts in the discipline and their comments are attached for authors consideration. Additionally, authors must consider below points;
1- problem statement that necessitated this study must be described in a single phrase at the start of abstract. Results should be stated objectively in abstract. Introduction section needs to be enriched by adding recent peer-findings in order to establish study rationale. Same goes for discussion section to support recorded findings. Sensory characteristics need further emphasis.

Reviewer 1 ·

Basic reporting

No comment

Experimental design

Research methodology is original, research question is well defined, up to date techniques were used and methodology was sufficient

Validity of the findings

no comment

Additional comments

This research article is about development of functional nuggets by supplementation of banana peel powder. No doubt fruit peels are rich in many minerals, vitamins, and important phytochemicals, also, vast literature is also available for apple, pomegranate, orange, etc. for their applications in functional food, this manuscript however, have some novel points such as, banana peel usage in chicken nuggets. Overall manuscript is well structured to deliver its objectives. This manuscript falls under the scope of this journal and present new knowledge for its readers. In my opinion, this manuscript is worthy for publication, however, there are few modifications/suggestions need to be incorporated before acceptance and publications.

Line 17-18: First sentence of the abstract need to be restructured.
Line 94: Heading “Development of Broiler Functional Nuggets should come after the heading “Compositional Study of Banana Peel Powder” at Line 105.
Line 107: describe drying/oven conditions.
Line 133: describe molarity of trichloroacetic acid used
Line 134: write full form of TBA
Line 153-155: Rewrite the sentence for clear understanding.
Line 156-161: write full form of all abbreviations, e.g. LSD, SPSS, SD, ANOVA etc.
Line 164-170: restructure the paragraph do not use too much “The” in every sentence, even in the whole manuscript. Use direct English for describe your expression.
Line 171-173: Please elaborate “Increased total phenolic content” from what, does authors make comparison with some standards or other studies?.
Line 192-193: statement is not clear. Rewrite the sentence, also present pH noted.
Line 194-194: Again its unclear that from which increase in phenolic content was observed?; Explain clearly what was compared in this statement, whether between treatments or from control.
Line 204-206: restructure the sentence
Line 210: At the end of these analysis, Author need to mention clearly, that what treatment at what day showed best results in terms of above analysis
Line 220-221: use uniform terminology of figure, in short of full form
Results of Sensory evaluation need to be elaborated more precisely.
Overall English language also need some improvements.
Figures graphic quality also need attention
Figure 6-9: Combine all four figures in one figure with clear a,b,c,d labeling and its relieving description
It is better to draw a flowchart/schematic representation of the whole procedure
Table-1 also need to be shorten to treatment / BPP % and meat quantity used. All other constant ingredients should me mention in the text or under the table as foot note.

Reviewer 2 ·

Basic reporting

.

Experimental design

.

Validity of the findings

.

Additional comments

Comments to Authors
Overall research work was presented in a good way to report the work on supplementation of banana peel powder for development of functional nuggets. There are some comments and suggestions for improvement of manuscript before finalization of the work.
Recommendation: Accept with minor revision
Line No
Comments
Abstract
Line 21
‘Refrigerator storage’ should be ‘storage at refrigeration temperature’
Line 26
Line 28

Line 31
Line 32
‘WHC’ should be written in full form as ‘Water Holding Capacity’
TBARS unit must be added for other value 0.14±0.02 mg malonadledhyde/Kg.
105 should be written as 105
“Treated with” should be replaced with “supplemented with”
Introduction
Line 46 Remove full stop before citation
Line 49 Change numerous “banana peels” To “enormous amount of banana”
peels
Materials and Methods
Line 86 & 87 Connect these lines by using “and”; write as “Banana peels were cut
into small pieces and treated with a 0.5% (w/v) solution of citric acid
for 10 minutes to prevent enzymatic browning”.
Line 89
Line 94
Line 107
The model and company of hot air oven should be mentioned?
Change Broiler Functional Nuggets to Functional Broiler Nuggets
Rephrase line as “crude fibre were determined as per standard protocols of AOAC (2005).
Line 113
Line 114
‘By taking the difference’ should be ‘using the formulae’
Add space before ash
Line 117
Line 117 & 118
Model and company of spectrophotometer should be mentioned
Combine both lines
Line 126
‘5oC’ should be corrected to ‘5oC’
Line 127
Line 140
Line 144
Which pH meter was used?
Change “ methods” To “method”
Change “loads” To “load” and remove article a before method of Salazar et al 2021.
Line 160
P-value should be mentioned as (p<0.05)
Results
Line 170-171


Line 171 & 178
Line 206
Line 213&214
Line 219

‘Total phenolic content was observed in banana peel powder were’ should be written as ‘Total phenolic content observed in banana peel powder was’
Add 0 before S.D
Change no incorporation of To 0% BPP
Remove S.D
Remove “for the fresh sample”
Line 222
Change “Colour” to “colour”
Discussion
Line 234
‘results by (Wachirasiri et al. 2009)’ should be written as ‘results observed by Wachirasiri et al. (2009)’
Line 262
‘water-holding capacity’ should be written without hyphen ‘water holding capacity’
Line 287
‘reduce a cooking loss’ should be written as ‘reduced cooking losses’
Line 300
Mention the product name
Conclusions
Line 310
Mention the storage time in days
Line 312
Line 313
Results contradict with the data provided in line 311
Change “5%” To “6%”

Reference list also needs author attention.

---

## Round 0.2 · accepted · Accept

The authors have carefully incorporated all suggestions forwarded by reviewers and editorial staff. The manuscript might be accepted in its present form now.

Reviewer 1 ·

Basic reporting

no comment

Experimental design

no comment

Validity of the findings

no comment

Additional comments

Authors have made all the suggested changes. now this manuscript can be accepted as it is.

Reviewer 2 ·

Basic reporting

No comment

Experimental design

No comment

Validity of the findings

No comment

Additional comments

No comment